# Association between low levels of HIV-1 DNA and HLA class I molecules in chronic HIV-1 infection

**Camilla Muccini** [1,2]*, **Monica Guffanti**[1], **Vincenzo Spagnuolo**[1,2], **Massimo Cernuschi**[1], **Laura Galli**[1], **Alba Bigoloni**[1], **Andrea Galli**[1], **Andrea Poli**[1], **Sara Racca**[3], **Antonella Castagna**[1,2]

**1** Infectious Diseases, IRCCS San Raffaele Scientific Institute, Milan, Italy, **2** Vita-Salute San Raffaele University, Milan, Italy, **3** Laboratory of Microbiology, IRCCS San Raffaele Scientific Institute, Milan, Italy

* muccini.camilla@hsr.it

**Data Availability Statement:** All relevant data are within the paper and its Supporting Information files.

## Abstract

### Background

HLA-B27 and -B57 were found in people with low levels of HIV-1 DNA, suggesting that HLA class I molecules may influence the size of HIV-1 reservoir. Aim of the study was to explore the association between HLA class I molecules and HIV-1 DNA in people with chronic HIV-1 infection.

### Methods

Post-hoc analysis of the APACHE trial, on adults with chronic HIV-1 infection, prolonged suppressive antiretroviral therapy and good immunological profile. HIV-1 DNA was quantified in peripheral blood mononuclear cells (PBMCs); HLA-A, -B and -C were tested on genomic DNA. Crude odds ratios (OR) with their respective 95% Wald confidence intervals (95% CIs) were estimated by univariable logistic regression for HLAs with a p-value <0.10.

### Results

We found 78 and 18 patients with HIV-1 DNA $\geq$100 copies/$10^6$PBMCs and with HIV-1 DNA <100 copies/$10^6$PBMCs, respectively. HLA-A24 was present in 21 (29.6%) participants among subjects with HIV-1 DNA $\geq$100 copies/$10^6$PBMCs and 1 (5.9%) among those with HIV-1 DNA <100 copies/$10^6$PBMCs (OR = 5.67, 95%CI = 0.79–46.03; p = 0.105); HLA-B39 was present in 1 (1.4%) with HIV-1 DNA $\geq$100 copies/$10^6$PBMCs and in 3 (17.6%) with HIV-1 DNA <100 copies/$10^6$PBMCs (OR = 13.71, 95%CI = 1.33–141.77; p = 0.028) and HLA-B55 in 3 (4.2%) and 3 (17.6%), respectively (OR = 4.43, 95%CI = 0.81–24.29; p = 0.087). All the three patients with HLA-B39 and HIV-1 DNA <100 copies/$10^6$PBMCs did not have HLA-A24.

### Conclusions

In patients with HIV-1 infection who maintained a good virological and immunological profile, HLA-B39 and -B55 may be associated with lower levels of HIV-1 DNA.

**Funding:** The authors received no specific funding for this work.

**Competing interests:** The authors have declared that no competing interests exist.

## Introduction

Over the past years, the role of host genetic factors in HIV-1 replication and disease progression to acquired immune deficiency syndrome (AIDS) has been widely investigated [1, 2].

In particular, human leukocyte antigen (HLA) is a group of highly polymorphic cell-surface proteins that play a key role in the regulation of immune system, presenting peptide antigens to cytotoxic T lymphocytes (CTLs) to eliminate the infected cells [3]. However, mechanisms of HIV-1 escape from CTL pressure have been previously reported [4].

Among major histocompatibility complex molecules, HLA class I alleles (-A, -B and -C) have demonstrated to be more significantly involved in the HIV-1 pathogenesis; especially, HLA-B appears to have the strongest influence on HIV outcome [5].

Therefore, several studies have focused attention on HLA class I profile of long term non progressors (LTNPs) to better identify determinants related to the spontaneous control of viral replication; nowadays, it is well documented that HLA-B27 and -B57 are associated with a slower HIV-1 disease progression [6, 7], while HLA-B35 with a poor prognosis [8].

Moreover, HLA-B27 and -B57 were also found in LTNPs and elite controllers with low levels of proviral HIV-1 DNA [9, 10], suggesting that HLA class I molecules may have an impact on the size of latent HIV-1 reservoir. Furthermore, data from four analytic treatment interruption trials have revealed an association between HLA-B alleles, including HLA-B27 and -B57, and a delayed viral rebound [11], emphasizing the role of cellular host immunity in controlling HIV-1 replication. These findings are raising interest since one of the greatest challenges towards a HIV cure is to characterize the viral reservoir.

Aim of the study was to explore if there were HLA class I alleles associated with low HIV-1 DNA in people living with HIV-1 (PLWH) chronic infection, prolonged suppressive antiretroviral therapy (ART) and good immunological profile.

## Methods

This is a post-hoc exploratory analysis of the APACHE trial, conducted at Infectious Diseases Clinic of the San Raffaele Hospital, Milan, Italy. The APACHE study is a prospective, open-label, single-arm, non-randomized, proof-of-concept study designed to characterize viral rebound during analytic treatment interruption. Subjects with chronic HIV-1 infection, HIV-1 RNA <50 copies/mL for ≥10 years, absence of plasma residual viremia for ≥5 years without any viral blips and CD4+ >500 cells/µL were screened for HIV-1 DNA [12]; long-term non-progressors and elite controllers were excluded. The study protocol was in accordance with the Declaration of Helsinki.

The Ethical Committe of the San Raffaele Hospital approved the study protocol (on 17/05/2016; approval reference number: 31/2016) and all the enrolled patients provided written informed consent.

The APACHE study is registered with ClinicalTrials.gov (NCT03198325, first posted on 26/06/2017).

HIV-1 RNA was quantified by kinetic polymerase chain reaction (PCR) molecular system (Abbott real-time PCR). Undetectable viral load was defined as HIV-1 RNA <50 copies/mL, residual viremia as detectable values of HIV-1 RNA <50 copies/mL (semi-quantitative result), while absence of residual viremia as the lack of detectable HIV-1 RNA in a sample (qualitative result).

At screening, total HIV-1 DNA was amplified as formerly described and quantified in peripheral blood mononuclear cells (PBMCs) by Real Time PCR (ABI Prism 7900) [13]; low HIV-1 DNA was established as HIV-1 DNA <100 copies/$10^6$ PBMCs.

At HIV-1 DNA testing, HLA-A, -B and -C were tested on genomic DNA using a PCR sequence-specific oligonucleotide (HISTO SPOT SSO) and a PCR sequence-specific primer (Olerup SSP kits).

All tests were performed in the Laboratory of Microbiology and Virology of IRCCS San Raffaele Scientific Institute.

Patients' characteristics were reported as median (quartile1-quartile3) or frequency (%). Baseline characteristics were compared using the chi-square test/Fisher's exact test or the Wilcoxon rank-sum test.

Crude odds ratios (OR) with their respective 95% Wald confidence intervals (95% CIs) were also estimated by univariable logistic regression for HLAs with a p-value <0.10.

All statistical tests were two-sided at the 5% level and were performed using SAS software (version 9.4; SAS Institute, Cary, NC).

## Results

At the time of enrolment, PLWH followed at San Raffaele Hospital were 4921: 99 met all the eligibility criteria of the APACHE study and 96 were screened for HIV-1 DNA. Overall, 78 (81%) had HIV-1 DNA $\geq$100 copies/$10^6$PBMCs and 18 (19%) <100 copies/$10^6$PBMCs.

At HIV-1 DNA testing, median age was 32 (25.2–38.9), 61 (64%) were male and 15 (15.6%) had a diagnosis of AIDS; furthermore, ART was started 7.9 (2.1–47.8) months after HIV diagnosis and received for a median time of 18.1 (15.6–19.8) years. HIV-1 RNA was <50 copies/mL for a median time of 11.7 (10.7–14.0) years and absence of plasma residual viremia for 6.9 (6.1–7.2) years. Current CD4+ count was 763 (605–922) cells/μL and CD4 +/CD8+ ratio 1.35 (0.86–2.02), while nadir CD4 T cell count was 253 (167–339) cells/μL among patients with HIV-1 DNA $\geq$100 copies/$10^6$PBMCs and 353 (212–434) cells/μL among those with HIV-1 DNA <100 copies/$10^6$PBMCs (p = 0.055). Overall, 66 (68.8%) PLWH had a previous virological failure: in more detail, 57 (73.1%) in people with HIV-1 DNA $\geq$100 copies/$10^6$PBMCs and 9 (50%) in people with HIV-1 DNA <100 copies/ $10^6$PBMCs (p = 0.088). The most frequent treatment regimen was a non-nucleoside reverse transcriptase inhibitor + 2 nucleoside reverse transcriptase inhibitors in both groups [in 34 (43.6%) and 8 (44.4%), respectively].

Other patients' characteristics are described in Table 1.

Among 96 participants screened for HIV-1 DNA, 88 were tested for HLA class I profile: we identified 71 (81%) patients with HIV-1 DNA $\geq$100 copies/$10^6$PBMCs and 17 (19%) with HIV-1 DNA <100 copies/$10^6$PBMCs.

At HIV-1 DNA determination, HLA class I molecules more frequently observed in subjects with HIV-1 DNA <100 copies/$10^6$PBMCs were HLA-A2, described in 11 (64.7%) participants, HLA-A11, -C3, -C6 and -C12, each represented in 4 (23.5%) participants and HLA-C7 in 8 (47.1%), as shown in Fig 1.

Overall, only 3 HLA class I molecules showed at least a marginal association with HIV-1 DNA: HLA-A24 was present in 21 (29.6%) participants among subjects with HIV-1 DNA $\geq$100 copies/$10^6$PBMCs and 1 (5.9%) among those with HIV-1 DNA <100 copies/$10^6$PBMCs (OR = 5.67, 95%CI = 0.79–46.03; p = 0.105); HLA-B39 in 1 (1.4%) with HIV-1 DNA $\geq$100 copies/$10^6$PBMCs and in 3 (17.6%) with HIV-1 DNA <100 copies/$10^6$PBMCs (OR = 13.71, 95%CI = 1.33–141.77; p = 0.028) and HLA-B55 in 3 (4.2%) and 3 (17.6%), respectively (OR = 4.43, 95%CI = 0.81–24.29; p = 0.087).

All the three patients with HLA-B39 and HIV-1 DNA <100 copies/$10^6$PBMCs did not have HLA-A24.

**Table 1. Participants' characteristics at HIV-1 DNA testing.**

| Characteristic | Category | HIV-1 DNA ≥100 copies/10⁶PBMCs (n = 78) | HIV-1 DNA <100 copies/10⁶PBMCs (n = 18) | p-value§ |
|---|---|---|---|---|
| Age at HIV diagnosis (years) | | 31.7 (25.2–37.7) | 34.0 (27.9–40.3) | 0.426[a] |
| Male gender | | 47 (60%) | 14 (78%) | 0.187[b] |
| AIDS diagnosis | No | 65 (83.3%) | 16 (88.9%) | 0.729[b] |
| | Yes | 13 (16.7%) | 2 (11.1%) | |
| HIV risk factor | Heterosexual | 39 (50%) | 6 (33%) | 0.034[b] |
| | MSM | 18 (23%) | 3 (17%) | |
| | PWID | 10 (13%) | 0 | |
| | Other | 11 (14%) | 9 (50%) | |
| HCV coinfection | No | 54 (69.2%) | 16 (88.9%) | 0.224[b] |
| | Yes | 21 (27%) | 2 (11.1%) | |
| | Unknown | 3 (3.8%) | 0 | |
| Nadir CD4+ (cells/μL) | | 253 (167–339) | 353 (212–434) | 0.055[a] |
| Zenith HIV-1 RNA (copies/mL) | | 51000 (11000–137708) | 31078 (7200–81919) | 0.542[a] |
| Year of ART start | | 1998 (1996–2000) | 1999 (1997–2002) | 0.194[a] |
| Months to ART start | | 12.0 (1.9–51.5) | 6.0 (2.8–24.1) | 0.526[a] |
| ART duration (years) | | 18.4 (15.7–20.2) | 17.5 (14.5–19.0) | 0.233[a] |
| Years of HIV-1 RNA <50 copies/mL | | 11.6 (10.6–13.5) | 12.3 (10.9–15.2) | 0.210[a] |
| Years of HIV-1 RNA <1 copy/mL | | 6.9 (6.1–7.1) | 6.8 (6.4–7.5) | 0.649[a] |
| HIV-1 DNA (copies/10⁶PBMC) | | 1100 (230–2475) | - | NA |
| CD4+ cell count (cells/μL) | | 826 (575–949) | 744 (660–773) | 0.884[a] |
| CD4+% | | 41.4 (32.2–50.8) | 37.8 (26.7–47.6) | 0.602[a] |
| CD8+ cell count (cells/μL) | | 632 (533–772) | 523 (377–1230) | 0.937[a] |
| CD8+% | | 30.1 (25.8–40.5) | 28.6 (23.6–41) | 0.692[a] |
| CD4+/CD8+ ratio | | 1.38 (0.93–2.07) | 1.35 (0.62–2.02) | 0.579[a] |

Results described by use of median (IQR) or frequency (%).

§ p-values by Wilcoxon rank-sum test (a) or chi-square/Fisher's exact test (b).

Abbreviations: PBMC, peripheral blood mononuclear cells; AIDS, acquired immunodeficiency syndrome; MSM, men who have sex with men; PWID, people who inject drugs; ART, antiretroviral therapy; NA, not applicable.

## Discussion

We have focused our attention on HLA class I molecules, in order to explore if they may be associated with a low peripheral viral reservoir in subjects with HIV-1 chronic infection.

Our findings suggest that in particular HLA-B39 and -B55 might be associated with HIV-1 DNA <100 copies/10⁶PBMCs. HLA-B39 was formerly found to be associated with LTNPs in a Spanish cohort [14] and with a lower viremia among PLWH in Zambia [15], consistently with our findings; therefore, both the studies have confirmed the influence of HLA-B39 on the control of viral replication.

However, HLA-B39 antigen was also previously classified as a risk allele due to a higher frequency described in seropositive compared to seronegative subjects, suggesting a role in the susceptibility to HIV-1 infection [16, 17].

For the first time, our analysis has mentioned HLA-B55, a split antigen from the B22 broad antigen, among the alleles detected in PLWH with a long-lasting virological suppression. The protective effect of HLA-B55 was already investigated in a study conducted in Argentina aiming to evaluate allele prevalence in HIV-1 positive people; this HLA allele was absent in HIV-1 infected patients, but present in control subjects [16]. However, HLA-B55, together with -B56, has also demonstrated to predispose to high levels of viremia and a faster progression to AIDS

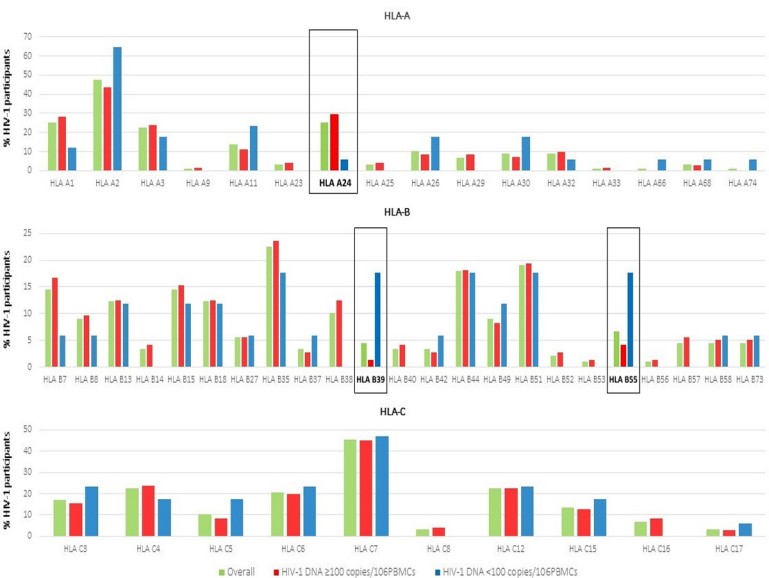

**Fig 1.** HLA class I (HLA-A, -B and–C) alleles are reported in the Fig 1 according to HIV-1 DNA values. HLA class I alleles more frequently observed in subjects with HIV-1 DNA <100 copies/10⁶PBMCs were HLA-A2, -A11, -C3, -C6, -C7 and -C12.

both in European and North American HIV-1 populations [18, 19]; multicenter studies conducting on a greater number of participants should be designed to clarify the role of HLA-B55 in HIV-1 infection.

Among individuals with HIV-1 DNA ≥100 copies/10⁶PBMCs, we observed a significant association with HLA-A24 that was present in nearly a third of PLWH with a higher viral reservoir.

In line with our data, the Multicenter AIDS Cohort Study showed that HLA-A24 was frequent in rapid CD4+ cells decliners [20], supposing that it might be a factor involved in HIV-1 disease progression; as evidence of the unfavorable impact of HLA-A24 antigen, it is rare in LTNPs [14].

The only association reported in literature between low HIV-1 DNA levels and HLA alleles was seen in LTNPs and elite controllers with HLA-B27 and -B57 [9, 10]. In our analysis, these protective alleles were unexpectedly described with a low frequency among people with HIV-1 DNA <100 copies/10⁶PBMCs and we did not find any correlation with the size of HIV-1 reservoir; further investigations on a larger sample size will be helpful to better interpret these findings.

In addition, discordant results from studies may be caused by different HIV-1 positive subjects' characteristics; in fact, recent data has revealed the importance of performing genome-wide association study to identify ethnical disparities in the host control of HIV-1 infection [21, 22].

One of the limitations of the study is the retrospective design that cannot exclude the persistence of residual confounding. Moreover, the small number of available patients, due to the restricted inclusion criteria applied in the APACHE study, limited the statistical power of the study and the generalizability of our findings. Nevertheless, the major strength relies on the availability of HIV-1 DNA values and then the opportunity to assess the original association between HLA class I alleles and a low HIV-1 peripheral reservoir.

In conclusion, in people with chronic HIV-1 infection, prolonged suppressive ART, absence of plasma residual viremia for ≥5 years and good immunological profile, the presence

of HLA-B39 and -B55 may be associated with a lower level of total HIV-1 DNA, highlighting the need to deeply evaluate the role of HLA class I in affecting the size of reservoir in further larger studies.

## Supporting information

**S1 File.**
(DOCX)

**S1 Data.**
(CSV)

## Acknowledgments

Thanks to all patients who participated to this study, nurses and physicians of the Infectious Diseases Clinic of the IRCCS San Raffaele Scientific Institute.

## Author Contributions

**Conceptualization:** Camilla Muccini.

**Data curation:** Laura Galli, Andrea Galli, Andrea Poli.

**Formal analysis:** Laura Galli, Andrea Poli.

**Investigation:** Monica Guffanti, Vincenzo Spagnuolo, Massimo Cernuschi.

**Methodology:** Laura Galli, Andrea Poli.

**Resources:** Alba Bigoloni, Sara Racca.

**Supervision:** Antonella Castagna.

**Validation:** Sara Racca.

**Visualization:** Alba Bigoloni, Andrea Galli.

**Writing – original draft:** Camilla Muccini.

**Writing – review & editing:** Monica Guffanti, Vincenzo Spagnuolo, Massimo Cernuschi, Antonella Castagna.

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
