## [Decision Letter · Decision Letter 0]

29 Dec 2021

PONE-D-21-31418Association between low levels of HIV-1 DNA and HLA class I molecules in chronic HIV-1 infectionPLOS ONE

Dear Dr. Muccini,

Thank you for submitting your manuscript to PLOS ONE. After careful consideration, we feel that it has merit but does not fully meet PLOS ONE’s publication criteria as it currently stands. Therefore, we invite you to submit a revised version of the manuscript that addresses the points raised during the review process.

You will see that the Referees found your work of some interest. However, they also raised major criticisms. Please respond to all the comments by Reviewers, with special attention to methodological points raised by Reviewer #1.

We look forward to receiving your revised manuscript.

Kind regards,

Giuseppe Vittorio De Socio, MD, PhD

Academic Editor

PLOS ONE

Journal Requirements:

Reviewers' comments:

Reviewer's Responses to Questions

**Comments to the Author**

1. Is the manuscript technically sound, and do the data support the conclusions?

Reviewer #1: No

Reviewer #2: Yes

Reviewer #3: Yes

2. Has the statistical analysis been performed appropriately and rigorously? 

Reviewer #1: No

Reviewer #2: Yes

Reviewer #3: Yes

3. Have the authors made all data underlying the findings in their manuscript fully available?

Reviewer #1: No

Reviewer #2: No

Reviewer #3: Yes

4. Is the manuscript presented in an intelligible fashion and written in standard English?

Reviewer #1: Yes

Reviewer #2: Yes

Reviewer #3: Yes

5. Review Comments to the Author

Reviewer #1: 1) The three reported signals are 21/78 vs. 1/18 (p=0.105), 1/78 vs. 3/16 (p=0.028), and 3/78 vs. 3/18 (p=0.087). If consider multiple testing adjustment, none is significant. Overall, there is no statistical power the declare any significant signal for the current study. More data or confirmation studies are needed.

Reviewer #2: This aimed at exploring the correlation between low levels of HIV-1 DNA and specific HLA alleles. Although results should be confirmed in a larger sample size, the study is interesting and clearly reported and discussed. However there are some minor points to be addressed:

- Please, report in more details the methodology used to assess the quantification of HIV-DNA and of residual viremia. This part can be also included in the supplementary material.

- Please, specify if the assay allows to quantify total or integrated HIV-DNA.

- Please, better specify the sentence “HIV-1 RNA was <50 copies/mL for a median time of 11.7 (10.6-14.0) years and absence of plasma residual viremia for 6.9 (6.2-7.2) years”. Does it mean that HIV-1 RNA or residual viremia was persistently undetectable or absent throughout the median time reported, with no episodes of viral blips?

- Please, modify “Nadir CD4+ were” in “Nadir CD4+ T cell count was”

- Which was the selection criteria for the 88 patients screened for HLA?

- In the footnotes of Table 1, please report the statistical tests used to assess statistical significance.

- In Figure 1, please highlight those HLA alleles significantly associated with different levels of HIV-1 DNA.

- Is there a correlation between HLA and the outcome of analytical treatment interruption?

Reviewer #3: I suggest to better clarify some aspects:

- More information about study population should be useful: year of enrolment, type of regimens and line of ARV, if they ever failed… Moreover, since the discussion includes consideration about LTNP and elite controllers, it could be useful clarify if in this study population, this kind of patients are present or not (according to my understanding not)

- I suggest to write in the main text the descriptive variables for all the patients, and to leave the differences among the two populations only in the table 1.

- I suggest to clarify at which timepoint the HIVDNA test was performed: at time of enrolment in Apache study? It was repeated?

- The authors stated that HLA-B27 and B57 are well documented and associated to HIV outcomes. I suppose that their prevalence is described in Figure 1 (it is not readable, it is too small) but also a comment in the text should be added. E.g. if some associations has been found also with this two HLAs.

- In the discussion, when HLA-B55 is mentioned, some previous data of other cohorts (references 17,18) seem to be in contrast with the findings of this study. Please give a comment on this.

6. PLOS authors have the option to publish the peer review history of their article (what does this mean?). If published, this will include your full peer review and any attached files.

Reviewer #1: No

Reviewer #2: No

Reviewer #3: No

---

## [Author Response · Author response to Decision Letter 0]

15 Jan 2022

Reviewer #1: 1) The three reported signals are 21/78 vs. 1/18 (p=0.105), 1/78 vs. 3/16 (p=0.028), and 3/78 vs. 3/18 (p=0.087). If consider multiple testing adjustment, none is significant. Overall, there is no statistical power the declare any significant signal for the current study. More data or confirmation studies are needed.

The issue raised by the Reviewer is important and we agree on the fact that the study is not adequately powered to draw firm conclusions. However, not only fully-powered studies have been published in the scientific literature but also exploratory, proof-of-concept or pilot studies; despite these studies are characterized by a limited sample size, they have generally the merit to explore new scientific hypotheses. 

We believe that the study design, the methods applied, the wording of our manuscript is in line with the small sample size; in fact, we proposed our work as an exploratory study, we conducted univariate analyses only, we stated that HLA-B39 and -B55 might be associated with HIV-1 DNA <100 copies/106PBMCs, we acknowledged among the limitations the small number of available patients and we concluded that the presence of HLA-B39 and -B55 may be associated with a lower level of total HIV-1 DNA, highlighting the need to deeply evaluate the role of HLA class I in affecting the size of reservoir.

According to the Reviewer’s comment, we also included in the discussion two additional considerations on these issues.

Reviewer #2: This aimed at exploring the correlation between low levels of HIV-1 DNA and specific HLA alleles. Although results should be confirmed in a larger sample size, the study is interesting and clearly reported and discussed. However there are some minor points to be addressed:

- Please, report in more details the methodology used to assess the quantification of HIV-DNA and of residual viremia. This part can be also included in the supplementary material.

According to the Reviewer’s comment, we added more details in the method section on residual viremia quantification and we also included a specific reference for HIV-DNA quantification where the assay used for HIV-DNA is clearly explained (De Rossi A et al. Quantitative HIV-1 proviral DNA detection: a multicentre analysis. New Microbiol. 2010; 33:293-302). 

- Please, specify if the assay allows to quantify total or integrated HIV-DNA.

We specified in the method that we quantified total HIV-DNA.

- Please, better specify the sentence “HIV-1 RNA was <50 copies/mL for a median time of 11.7 (10.6-14.0) years and absence of plasma residual viremia for 6.9 (6.2-7.2) years”. Does it mean that HIV-1 RNA or residual viremia was persistently undetectable or absent throughout the median time reported, with no episodes of viral blips?

Thanks for the comment; we added in the method section “without any viral blips” to better clarify the inclusion criteria of the study.

- Please, modify “Nadir CD4+ were” in “Nadir CD4+ T cell count was”

Thanks for the suggestion, we changed “Nadir CD4+ were” in “Nadir CD4+ T cell count was” in the text.

- Which was the selection criteria for the 88 patients screened for HLA?

Among 96 participants screened for HIV DNA, we have available HLA data for 88 patients.

- In the footnotes of Table 1, please report the statistical tests used to assess statistical significance.

According to the Reviewer’s comment, Table 1 now includes the statistical test applied to compare the two groups of patients. 

- In Figure 1, please highlight those HLA alleles significantly associated with different levels of HIV-1 DNA.

We modified Figure 1 according to the Reviewer’s comment.

- Is there a correlation between HLA and the outcome of analytical treatment interruption?

We explored a possible correlation between HLA and the outcome of analytical treatment interruption but we did not find any association.

Reviewer #3: I suggest to better clarify some aspects:

- More information about study population should be useful: year of enrolment, type of regimens and line of ARV, if they ever failed… Moreover, since the discussion includes consideration about LTNP and elite controllers, it could be useful clarify if in this study population, this kind of patients are present or not (according to my understanding not)

According to the Reviewer’s comment, we added the requested information into the manuscript. The exploratory study here reported is a parent study of the APACHE trial that excluded LTNP and elite controllers (enrolled patients had to stop ART); so, we clarified this issue in the methods of the manuscript.

- I suggest to write in the main text the descriptive variables for all the patients, and to leave the differences among the two populations only in the table 1.

As suggested, we removed the overall column from the table and included in the text some overall patients’ characteristics. 

- I suggest to clarify at which timepoint the HIVDNA test was performed: at time of enrolment in Apache study? It was repeated?

As reported in the manuscript, HIV-DNA was performed to screen participants of the APACHE study; it was repeated only in the patients included in the APACHE study who underwent analytic treatment interruption (ATI). HIV-DNA trend during ATI remained below 100 copies/106PBMCs at all study visits except in two participants.

- The authors stated that HLA-B27 and B57 are well documented and associated to HIV outcomes. I suppose that their prevalence is described in Figure 1 (it is not readable, it is too small) but also a comment in the text should be added. E.g. if some associations has been found also with this two HLAs.

According to the Reviewer’s comment, we better clarified in the discussion section the role of HLA-B27 and HLA-B57 in our study (“In our analysis, these protective alleles were unexpectedly described with a low frequency among people with HIV-1 DNA <100 copies/106PBMCs and we did not find any correlation with the size of HIV-1 reservoir; further investigations on a larger sample size will be helpful to better interpret these findings.”)

Moreover, HLA-B27 was present in 5 participants (5.6%; 4 with HIV-DNA ≥100 copies/106PBMCs and 1 with HIV-DNA <100 copies/106PBMCs), while HLA-B57 in 4 (4.5%), all with HIV-DNA ≥100 copies/106PBMCs.

- In the discussion, when HLA-B55 is mentioned, some previous data of other cohorts (references 17,18) seem to be in contrast with the findings of this study. Please give a comment on this.

Thanks for the suggestion, we made a comment in the discussion section about these data (“multicenter studies conducting on a greater number of participants should be designed to clarify the role of HLA-B55 in HIV-1 infection”).

---

## [Decision Letter · Decision Letter 1]

1 Mar 2022

Association between low levels of HIV-1 DNA and HLA class I molecules in chronic HIV-1 infection

PONE-D-21-31418R1

Dear Dr. Muccini,

We’re pleased to inform you that your manuscript has been judged scientifically suitable for publication and will be formally accepted for publication once it meets all outstanding technical requirements.

Kind regards,

Giuseppe Vittorio De Socio, MD, PhD

Academic Editor

PLOS ONE

Additional Editor Comments (optional):

The paper is suitablle  for publication, as the concerns raised by the reviewer #1, namely that the study is underpowered and it doesn't add any value to clarify this controversial topic, did not undermine the conclusions drawn in the paper presented as an exploratory study.

Reviewers' comments:

Reviewer's Responses to Questions

**Comments to the Author**

1. If the authors have adequately addressed your comments raised in a previous round of review and you feel that this manuscript is now acceptable for publication, you may indicate that here to bypass the “Comments to the Author” section, enter your conflict of interest statement in the “Confidential to Editor” section, and submit your "Accept" recommendation.

Reviewer #1: All comments have been addressed

Reviewer #3: All comments have been addressed

2. Is the manuscript technically sound, and do the data support the conclusions?

Reviewer #1: No

Reviewer #3: Yes

3. Has the statistical analysis been performed appropriately and rigorously? 

Reviewer #1: Yes

Reviewer #3: Yes

4. Have the authors made all data underlying the findings in their manuscript fully available?

Reviewer #1: No

Reviewer #3: Yes

5. Is the manuscript presented in an intelligible fashion and written in standard English?

Reviewer #1: Yes

Reviewer #3: Yes

6. Review Comments to the Author

Reviewer #1: The relationship between HLA-B39 and rival replication is controversial in literature, as shown in the first paragraph of Discussion, and the current study doesn't add any value to clarify the controversy. If it were the first study to explore the relationship, it may have some values. Being aware of the controversy, one should design a study to solve the problem instead.

Reviewer #3: (No Response)

7. PLOS authors have the option to publish the peer review history of their article (what does this mean?). If published, this will include your full peer review and any attached files.

Reviewer #1: No

Reviewer #3: No

---

## [Editor Report · Acceptance letter]

7 Mar 2022

PONE-D-21-31418R1 

Association between low levels of HIV-1 DNA and HLA class I molecules in chronic HIV-1 infection 

Dear Dr. Muccini:

I'm pleased to inform you that your manuscript has been deemed suitable for publication in PLOS ONE. Congratulations! Your manuscript is now with our production department. 

Kind regards, 

on behalf of

Dr. Giuseppe Vittorio De Socio 

Academic Editor

PLOS ONE